# Synthesis and Characterization of Acid-Activated Carbon Prepared from Sugarcane Bagasse for Furfural Production in Aqueous Media

**Thiago Alves Lopes Silva** [1] , **Adilson Candido da Silva** [2] **and Daniel Pasquini** [1,*]

1    Chemistry Institute (IQ-UFU), Federal University of Uberlândia, Uberlândia 38400-902, Brazil;
     thiago_1209@hotmail.com
2    Department of Chemistry, Institute of Exact and Biological Sciences (ICEB), Federal University of Ouro
     Preto—UFOP, Ouro Preto 35400-000, Brazil; adilsonqui@ufop.edu.br
*    Correspondence: daniel.pasquini@ufu.br; Tel.: +55-34-3239-4434

**Abstract:** Furfural is a platform molecule obtained from hemicellulosic monosaccharides present in lignocellulosic biomass. Due to the possibility of converting this molecule into several value-added chemicals and the need to search for more sustainable production processes, the present work aimed to produce and characterize different sulfonated activated carbons (AC-S) from sugarcane bagasse (SCB) for application in furfural production in aqueous media. ACs were produced by chemical activation using salts of $ZnCl_2$, $NiCl_2$, and $CuCl_2$ and a temperature and activation time of 550 °C and 3 h under nitrogen flow, respectively. Sulfonation was carried out with $H_2SO_4$ (98%) at a solid/liquid ratio of 1:10 at 160 °C for 2 h. Catalytic tests were performed using 5% catalyst mass regarding xylose, a temperature of 180 °C, and a reaction time of 2 h. ACs with high surface areas, ranging from 290 to 1100 $m^2 \ g^{-1}$, were produced. All catalysts had an increased sulfur content and total acidity after sulfonation, indicating the successful attachment of the sulfonic group (-$SO_3H$) in the carbon matrix of the CAs. The AC-S/$CuCl_2$ catalyst achieved the best catalytic performance compared to AC-S/$ZnCl_2$, AC-S/$NiCl_2$, and other acidic solids reported in the literature, achieving yield and selectivity of 55.96% and 83.93%, respectively. These results evidence the importance of the synergy between the Lewis and Brønsted acid sites on selective xylose dehydration and make AC-S/$CuCl_2$ a promising acid catalyst for converting xylose to furfural in an aqueous medium.

**Keywords:** sulfonated activated carbon; heterogeneous catalyst; furfural

## 1. Introduction

Currently, lignocellulosic biomass (LCB) conversion into biofuels and platform molecules has attracted attention due to concerns about reducing fossil reserves and the environmental impacts caused by using non-renewable fuels [1–3]. LCB is mainly composed of cellulose (35–55%), hemicelluloses (20–40%), and lignin (15–25%) [4,5]. For the conversion of the three major components of LCB into fuels and platform molecules, a fractionation step is necessary to reduce their recalcitrance and make the catalytic conversion process more efficient [6,7].

Generally, after LCB fragmentation, the $C_6$ sugar fraction released from cellulose is mainly used to produce second-generation ethanol, while the $C_5$ sugar fraction of hemicelluloses is considered a raw material that is very important for the synthesis of valuable chemicals [8]. Among these chemicals synthesized from hemicellulosic monosaccharides (principally xylose), one of the most important is furfural (FF). FF is a renewable platform molecule that can be converted into several chemical inputs applied in the plastic industry, including fertilizers, pesticides, and pharmaceutical products. In addition, it is a precursor for different furanic compounds, such as furfuryl alcohol, tetrahydrofurfuryl alcohol, 1,5-pentanediol, tetrahydrofuran, 2-methyl furan, maleic anhydride, furoic acid, and levulinic acid [9–11].

The industrial production of furfural started in 1921 by the Quaker Oats Company, using oat hulls as raw material subjected to high-pressure steam in the presence of sulfuric acid ($H_2SO_4$) [12,13]. In the global market for FF, China is the largest producer, followed by the Dominican Republic and South Africa. These three countries represent 90% of the production capacity in the world [14,15].

The current technology used in the industrial production of FF is based on the hydrolysis of biomass (corn cobs, sugarcane bagasse, oat hulls, wheat straw, peanut shells, cottonseed hull bran, cornstalks, etc.) [16,17], followed by dehydration of pentoses using a homogeneous Brønsted acid catalyst (mineral acids) in an aqueous medium. The homogeneous catalysis process has disadvantages, such as low yield (only ~50% of the theoretical furfural yield), corrosion of the industrial equipment, high energy demand, difficulty in separating the catalyst from the reaction medium, and high solvent (e.g., water, alkali) consumption for the pre-treatment of solid and liquid waste generated in the FF production process [13,18,19]. Therefore, to overcome the technology bottlenecks in commercial production, it is necessary to search for more efficient and sustainable catalytic processes, which increase the economic viability of converting carbohydrates into FF and improve its competition with petroleum-derived chemicals [20–22].

In this scenario, the use of heterogeneous acid catalysts for the selective dehydration of xylose into FF has gained more and more attention due to their high catalytic activity, low corrosivity, good thermostability, and ease of recycling and separation of the reaction medium [23,24]. Different heterogeneous catalysts applied to FF production have been reported in the literature, including zeolites, sulfonic ion exchange resins, mesoporous silicas modified with sulfonic acid, metallic oxides, and mesoporous niobium phosphate [15,23,25,26], and, more recently, carbonaceous materials derived from LCB [26–31], such as sulfonated activated carbons [32–36], which have high surface area and porosity, low production cost, and ease of performing surface chemical modifications. However, the primary limitation in the utilization of these acidic solids lies in their susceptibility to deactivation in aqueous and/or biphasic systems. In this perspective, the use of acid solids, such as sulfonated activated carbon obtained from LCB, can be a promising alternative to make the FF production process more sustainable and competitive.

In this context, the present work aimed to produce and characterize different sulfonated activated carbons obtained from sugarcane bagasse using a chemical activation method with zinc chloride ($ZnCl_2$), copper chloride ($CuCl_2$), and nickel chloride ($NiCl_2$), as well as evaluate the application of these in the xylose dehydration reaction to obtain furfural. The main novelty of this study is carrying out the sulfonation process in activated carbon-containing metals remaining from the activation process, resulting in the single-step synthesis of heterogeneous catalysts with Lewis (metals) and Brønsted (sulfonic group) acid sites. To the best of our knowledge, this is the first approach reported to have produced AC-S without performing the leaching step of the activating agent before the direct functionalization with $H_2SO_4$. The elimination of this step contributes to reducing costs and making FF production more eco-friendly since it would be necessary to use hydrochloric acid (HCl) to remove metals and water to wash the catalyst until it reaches a neutral pH. In addition, the results of this study can contribute to obtaining more efficient and sustainable catalysts for furfural production in aqueous media, thus composing the portfolio of possibilities to produce this platform molecule.

## 2. Results and Discussion

### 2.1. Carbonization and Sulfonation Process Yields

Table 1 shows the yields of the carbonization and sulfonation processes.

The carbonization process yield using $ZnCl_2$ and $NiCl_2$ showed close values; however, they were lower than the yield obtained using $CuCl_2$ as an activating agent. The observed yields are in the range reported for producing activated carbons by the chemical activation method with different precursor materials and activating agents, according to studies by

Zhang et al. [37], Liew et al. [38], Norouzi et al. [39], Fu et al. [40], Benmahdi et al. [41], and Mbarki et al. [42].

**Table 1.** Carbonization and sulfonation yields of the AC.

| Sample | $Y_C$ (%) [a] | $Y_S$ (%) [b] |
|---|---|---|
| AC/NiCl$_2$ | 28.50 | 77.70 |
| AC/ZnCl$_2$ | 24.90 | 72.20 |
| AC/CuCl$_2$ | 38.90 | 53.20 |

[a] $Y_C$—carbonization yield. [b] $Y_S$—sulfonation yield.

AC/CuCl$_2$ exhibited a lower sulfonation yield when compared to AC/NiCl$_2$ and AC/ZnCl$_2$, likely due to the reaction of metallic copper and H$_2$SO$_4$ with the loose carbon atoms in macro and mesopores of the carbonic structure initially formed. This reaction possibly initiates a new transverse or radial activation, thus forming a new microporous structure that promotes an increase in S$_{BET}$ [43]. Furthermore, it leads to the degradation of the carbon matrix and the removal of a significant portion of the remaining activating agent, resulting in a reduction in yield.

## 2.2. Catalyst Characterization

### 2.2.1. Textural Properties

The specific surface area (S$_{BET}$), micropore area (S$_{micro}$), external area (S$_{meso}$), total pore volume (V$_{tot}$), micropore volume (V$_{micro}$), and average pore diameter (d$_p$) of the ACs before and after sulfonation are showed Table 2.

**Table 2.** Textural properties of AC and AC-S.

| Samples | $S_{BET}$ [a] $(m^2\,g^{-1})$ | $S_{micro}$ [b] $(m^2\,g^{-1})$ | $S_{meso}$ [c] $(m^2\,g^{-1})$ | $V_{tot}$ [d] $(cm^3\,g^{-1})$ | $V_{micro}$ [e] $(cm^3\,g^{-1})$ | $V_{meso}$ [f] $(cm^3\,g^{-1})$ | $d_p$ [g] (nm) | $S_{micro}/S_{BET}$ [h] (%) | $V_{micro}/V_{tot}$ [i] (%) |
|---|---|---|---|---|---|---|---|---|---|
| AC/NiCl$_2$ | 295.91 | 224.74 | 71.17 | 0.226 | 0.115 | 0.111 | 3.06 | 76 | 51 |
| AC-S/NiCl$_2$ | 291.83 | 246.31 | 45.52 | 0.200 | 0.140 | 0.060 | 2.74 | 84 | 70 |
| AC/ZnCl$_2$ | 1108.84 | 756.11 | 352.73 | 0.600 | 0.398 | 0.202 | 2.15 | 68 | 66 |
| AC-S/ZnCl$_2$ | 1055.62 | 729.97 | 325.65 | 0.607 | 0.440 | 0.167 | 2.30 | 69 | 72 |
| AC/CuCl$_2$ | 300.82 | 264.75 | 36.07 | 0.185 | 0.129 | 0.056 | 2.46 | 88 | 69 |
| AC-S/CuCl$_2$ | 440.00 | 383.22 | 56.78 | 0.318 | 0.212 | 0.106 | 2.89 | 87 | 66 |

[a] $S_{BET}$—specific surface area. [b] $S_{micro}$—t-Plot micropore area. [c] $S_{meso}$—t-Plot external area. [d] $V_{tot}$—total pore volume. [e] $V_{micro}$—t-Plot micropore volume. [f] $V_{meso} = V_{tot} - V_{micro}$. [g] $d_p$—average pore diameter. [h] Proportion of microporous surface area to total specific surface area. [i] Proportion of microporous volume to total volume.

AC/ZnCl$_2$ showed a specific surface area 3.6 times greater than those activated with NiCl$_2$ and CuCl$_2$. Obtaining AC with high S$_{BET}$ using ZnCl$_2$ is reported in the literature [44,45] and attributed to the swelling of the molecular structure of cellulose, which causes the breaking of its lateral bonds and promotes the increase of intra- and inter-fibril spaces [46,47].

Luo et al. [48], Cai et al. [49], Abdel-Aziz et al. [50], Piriya et al. [51], Shudi and Wang [52], Kumar et al. [53], and Amoo et al. [54] also synthesized AC using ZnCl$_2$ as an activating agent, however, obtained S$_{BET}$ smaller when compared with AC/ZnCl$_2$. The textural properties results of AC/ZnCl$_2$ and AC/NiCl$_2$ (Table 2) are similar to those described by Thue et al. [55], who used sapelli sawdust impregnated with ZnCl$_2$ and NiCl$_2$ in a mass ratio of 1:1, and also produced ACs with S$_{BET}$ = 1123 m$^2$ g$^{-1}$ and 297 m$^2$ g$^{-1}$; V$_{tot}$ = 0.54 cm$^3$ g$^{-1}$ and 0.22 cm$^3$ g$^{-1}$, and d$_p$ = 1.9 nm and 2.9 nm, respectively. Nonetheless, the AC produced with CuCl$_2$ (1:1 *w/w*) showed higher S$_{BET}$ = 648 m$^2$ g$^{-1}$ and V$_{tot}$ = 0.36 cm$^3$ g$^{-1}$ than those of this work, possibly due to the differences in the carbonization process and of the physicochemical characteristics of the precursor.

After sulfonation, there is a reduction in S$_{BET}$, V$_{tot}$, and d$_p$ for AC-S/NiCl$_2$ (Table 2), which suggests the insertion of sulfonic groups (-SO$_3$H) on the surface and inside the pores [56]. While AC-S/ZnCl$_2$ showed a reduction only in S$_{BET}$. AC-S/CuCl$_2$ had a

significant increase in $S_{BET}$, $V_{tot}$, and $d_p$, which can be attributed to a new activation of AC promoted for the reaction of metallic copper and $H_2SO_4$, as described in Section 2.1.

The $S_{BET}$ reduction in carbonaceous materials after sulfonation was also reported in the studies by Niu et al. [57], Kolur et al. [58], Ferreira et al. [59], Higai et al. [60], and Zhang et al. [61], who related this reduction with the fixation of -$SO_3H$ on the surface and the degradation of the carbonic matrix by $H_2SO_4$.

Figure 1 shows the isotherms of adsorption and desorption of $N_2$ for the activated carbons before and after sulfonation.

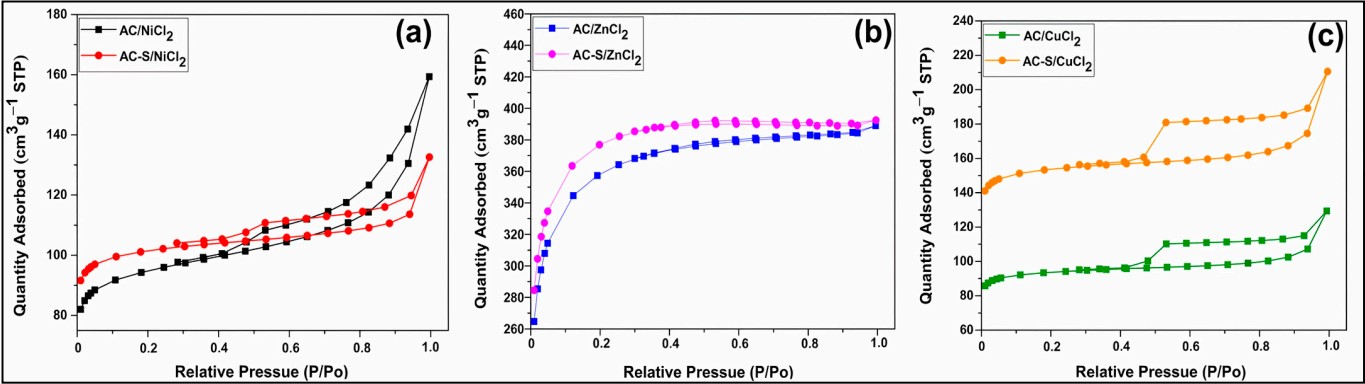

**Figure 1.** $N_2$ adsorption/desorption isotherms for (**a**) AC/NiCl$_2$ and AC-S/NiCl$_2$, (**b**) AC/ZnCl$_2$ and AC-S/ZnCl$_2$, (**c**) AC/CuCl$_2$ and AC-S/CuCl$_2$.

According to the IUPAC classification, the AC/ZnCl$_2$ and AC-S/ZnCl$_2$ (Figure 1b) have a type I(b) isotherm characteristic of microporous materials with wider micropores and possibly narrow mesopores. At the same time, AC/NiCl$_2$, AC-S/NiCl$_2$ (Figure 1a), AC/CuCl$_2$, and AC-S/CuCl$_2$ (Figure 1c) exhibit a type IV(a) isotherm, which is characteristic of mesoporous materials. The mesoporosity of AC/NiCl$_2$, AC-S/NiCl$_2$, is confirmed by the H3 hysteresis loop (before sulfonation) and H4 (after sulfonation), which also indicates a change from slit-shaped pores to narrow slit-shaped pores. AC/CuCl$_2$ and AC-S/CuCl$_2$ showed an H4 hysteresis loop, showing that sulfonation did not change the structure of the pores, which remained narrow in the form of a slit [62,63].

### 2.2.2. SEM-EDX Analysis

Figure 2 shows scanning electron microscopy and EDX composition obtained for the catalysts before and after functionalization. The porous structure of the AC$_S$ synthesized in this work is similar to other activated carbons produced with different carbon sources [64–67]. It is observed in micrographs of CA/NiCl$_2$ (Figure 2a) and CA-S/NiCl$_2$ (Figure 2d), which, after reaction with $H_2SO_4$, showed cracks and peeling of the internal layer of the pores (red arrows inserted in Figure 2d) due to partial degradation of the carbonic structure. The EDX maps show agglomerated nickel particles inside the pores (Figure 2a), but after sulfonation, the nickel was dispersed homogeneously on the sample surface (Figure 2d).

Figure 2e evidence that after sulfonation, there were no changes in the pore structure of AC/ZnCl$_2$ (Figure 2b), but the zinc particles (blue arrows inserted in Figure 2b) from on the surface and interior of the pores (Figure 2b) were removed. The micrographs of AC/CuCl$_2$ (Figure 2c) and AC-S/CuCl$_2$ (Figure 2f) demonstrated the formation of a spongy structure inside some pores of the sulfonated sample (green arrows inserted in Figure 2f).

Comparing EDX maps in Figure 2c,f a high leaching of copper after treatment with $H_2SO_4$ is observed. The presence of the sulfur (S) element in all sulfonated samples (Figure 2d–f) suggests the insertion of -$SOH_3$ groups in the matrix carbonic.

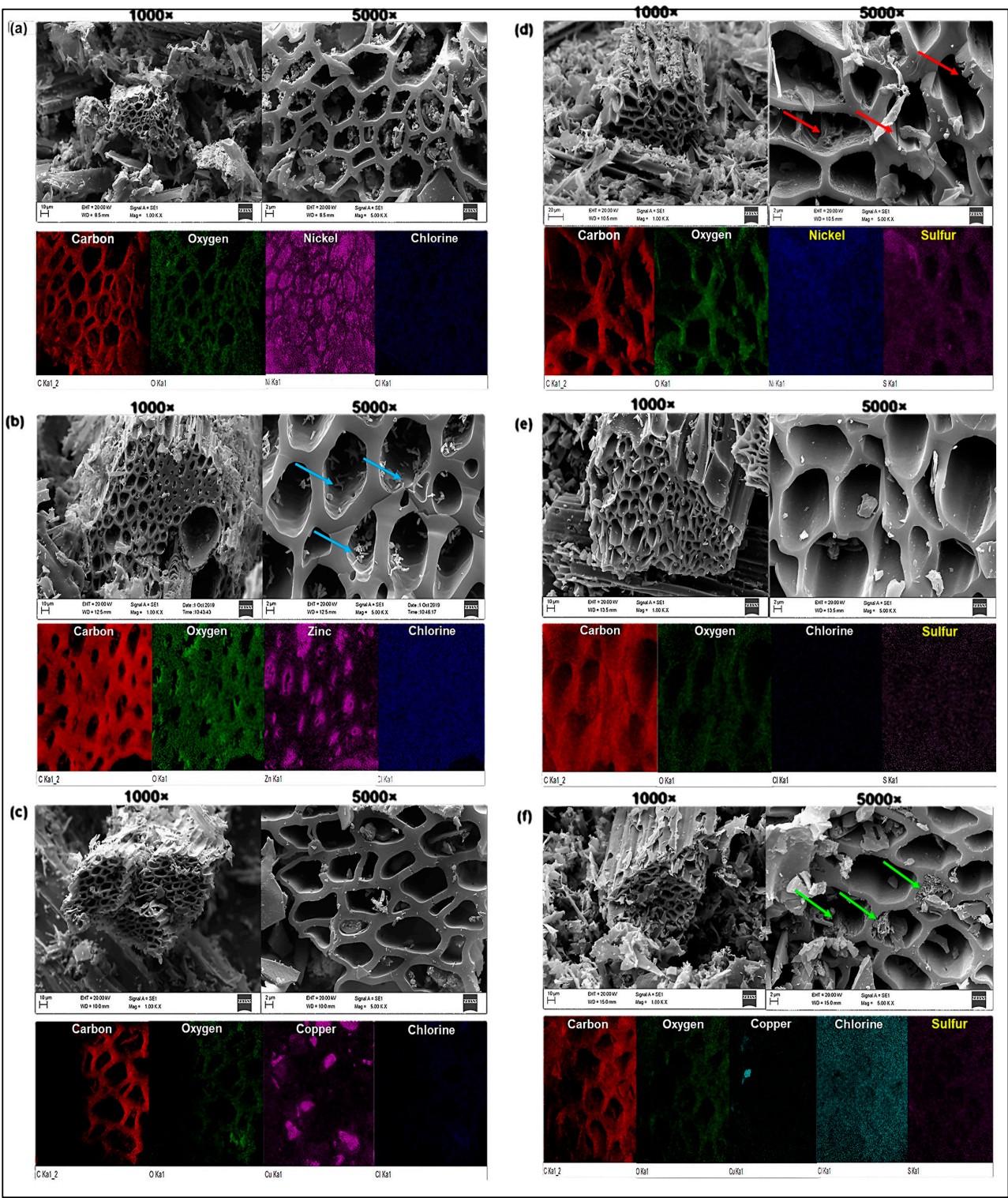

**Figure 2.** Micrographs and EDX composition for (**a**) AC/NiCl2, (**b**) AC/ZnCl2 containing zinc particles (blue arrows) on the surface and inside the pores, (**c**) AC/CuCl2, (**d**) AC-S/NiCl2 with cracks and peeling of the internal layer of the pores (red arrows), (**e**) AC-S/ZnCl2, and (**f**) AC-S/CuCl2 showing the formation of a spongy structure inside some pores (green arrows).

### 2.2.3. Elementary Analysis and Ion Exchange Capacity

Table 3 describes the results of the elemental chemical composition and total acidity for AC and AC-S samples. The sulfur content for all samples after the sulfonation process increased (Table 4), corroborating with the composition determined by EDX described in

Section 2.2.2 and, thus, confirming the anchoring of -SO$_3$H groups in the AC structure. The relationship between the increase in S content after sulfonation and the insertion of -SO$_3$H groups observed in this study has been extensively reported in several works on the synthesis and characterization of acid solid carbon catalysts [30,33,68–74]. These studies highlight that the sulfur atoms added to the carbon matrix of the catalyst after sulfonation are likely contained in -SO$_3$H groups generated and covalently linked by sp$_2$ hybridization in the polycyclic carbon network.

**Table 3.** Elemental chemical composition and acidity total of the AC and AC-S.

| Samples | %C | %H | %N | %S | %O | %Si * | % Cl * | %Ni * | %Zn * | %Cu * | Total Acidity mmol g$^{-1}$ | -SO$_3$H mmol g$^{-1}$ |
|---|---|---|---|---|---|---|---|---|---|---|---|---|
| AC/NiCl$_2$ | 47.81 | 1.06 | 0.19 | 0.21 | 29.23 | - | 0.16 | 21.34 | - | - | 0.02 ± 0.001 | - |
| AC-S/NiCl$_2$ | 52.52 | 1.94 | 0.23 | 3.63 | 27.46 | 0.15 | - | 14.07 | - | - | 0.30 ± 0.001 | 0.28 ± 0.001 |
| AC/ZnCl$_2$ | 65.40 | 1.02 | 0.30 | 0.19 | 26.35 | 0.19 | 0.49 | - | 6.06 | - | 0.04 ± 0.002 | - |
| AC-S/ZnCl$_2$ | 61.03 | 1.25 | 0.40 | 0.54 | 36.29 | 0.22 | 0.27 | - | - | - | 0.09 ± 0.002 | 0.05 ± 0.002 |
| AC/CuCl$_2$ | 41.40 | 0.76 | 0.23 | 0.13 | 38.75 | - | 0.51 | - | - | 18.22 | N.D. | - |
| AC-S/CuCl$_2$ | 63.71 | 2.10 | 0.42 | 1.94 | 29.22 | 0.39 | 0.44 | - | - | 1.78 | 0.33 ± 0.001 | 0.33 ± 0.001 |

* Determination by EDX. N.D.—not detect.

**Table 4.** Catalytic performance of acid solids catalyst in dehydration xylose for furfural production.

| Samples | C (%) | Y (%) | S (%) |
|---|---|---|---|
| No catalyst | 48.84 | 28.75 | 58.86 |
| AC/NiCl$_2$ | 75.00 | 22.72 | 30.79 |
| AC-S/NiCl$_2$ | 89.15 | 48.63 | 54.54 |
| AC/ZnCl$_2$ | 90.00 | 10.00 | 11.11 |
| AC-S/ZnCl$_2$ | 71.43 | 31.27 | 43.00 |
| AC/CuCl$_2$ | 80.00 | 35.67 | 44.60 |
| AC-S/CuCl$_2$ | 66.67 | 55.96 | 83.93 |

Analyzing the results of total acidity and concentration of -SO$_3$H groups in the sulfonated catalysts (Table 3), it was observed that AC-S/CuCl$_2$ had the highest concentration of these groups, followed by AC-S/NiCl$_2$ and AC-S/ZnCl$_2$. The lower acidity and insertion of -SO$_3$H groups in AC-S/NiCl$_2$, even presenting higher S content compared to AC-S/CuCl$_2$, indicates that some sulfonic groups of this catalyst have some steric impediment that made them unavailable for ion exchange reaction.

2.2.4. TGA Analysis

Thermogravimetric curves obtained in an inert atmosphere of the AC and AC-S are shown in Figure 3. The thermograms indicate that the AC-S initiates the thermal degradation of the carbonic structure at a temperature of approximately 200–220 °C. After sulfonation, all AC had their thermal stability reduced and a greater loss mass between 25–100 °C, indicating that these materials present more adsorbed water. The reduction in the thermal stability observed possibly occurred due to partial oxidation of the carbon structure caused by the sulfonation process [75], while the increase in the percentage of adsorbed water is a result of the insertion of -SO$_3$H groups, which increase the hydrophilicity of AC [76].

In the TGA profile of AC-S/NiCl$_2$ (Figure 3a) and CA-S/CuCl$_2$ (Figure 3c), three thermal decomposition events are observed, the first (25 to 100 °C) attributed to the vaporization of the water physically adsorbed in the sample [76]. The second (100 to 450 °C) refers to the decomposition of the surface groups present in the AC, such as the carboxylic groups (100 to 200 °C), lactones groups (200 to 400 °C) [77], and sulphonic groups (200 to 450 °C) [78–80]. The third event (450 to 900 °C) is attributed to the decomposition of the carbonic structure. In the AC-S/ZnCl$_2$ thermogram (Figure 3b), the occurrence of only

two thermal events is observed; the first stage from 25 to 100 °C was due to the release of adsorbed water, and the second from 100 to 900 °C to the thermal decomposition of the carbonic matrix. The desulfonation process in this AC-S is possibly not observed due to the low incorporation of sulfonic groups (according to the result shown in Table 3) compared to the other sulfonated catalysts.

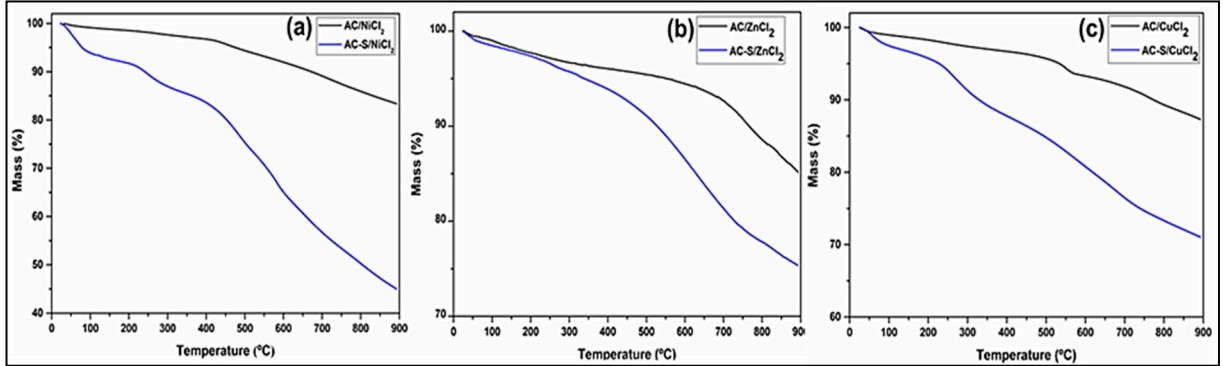

**Figure 3.** Thermogravimetric curves for (**a**) AC/NiCl$_2$ and AC-S/NiCl$_2$, (**b**) AC/ZnCl$_2$ and AC-S/ZnCl$_2$, (**c**) AC/CuCl$_2$ and AC-S/CuCl$_2$.

### 2.2.5. XRD and Raman Analysis

Figure 4a shows the XRD patterns of the activated carbons before and after sulfonation. The diffractograms of the AC/NiCl$_2$, AC/ZnCl$_2$, and AC/CuCl$_2$ present, respectively, characteristic peaks of metallic nickel (Ni$^0$, ICCD card 00-004-0850) at 2θ ≈ 44.50° (111), 51.84° (200), and 76.36° (220); zinc oxide (ZnO, ICCD card 00-036-1451) at 2θ ≈ 31.77° (100), 34.42° (002), 36.25° (101), 47.54° (102), 56.60° (110), 62.86° (103), 66.38° (200), and 69.10° (201); metallic copper (Cu$^0$, ICCD card 00-004-0836) at 2θ ≈ 43.29° (111), 50.4° (200), and 74.13° (220). According to Thue et al. [81], the formation of Ni$^0$, ZnO, and Cu$^0$ is possibly associated with the occurrence of three main stages during the production of AC: the first is the impregnation of the activating agent in the biomass, which promotes the complexation of the metallic cation with the surface groups of cellulose, hemicelluloses (-OH, -COOH), and lignin (Ph-OH). The second consists of the decomposition of biomass during the carbonization process, in which gases are released, such as H$_2$, CO$_2$, CO, and CH$_4$. Finally, in the third stage, reducing gases react with metallic cations at high temperatures, promoting the reduction of Ni$^{2+}$ and Cu$^{2+}$ to Ni$^0$ and Cu$^0$ particles, while Zn$^{2+}$ forms ZnO particles. The presence of ZnO observed in the diffractograms is in line with the results related by Ngaosuwan et al. [82] and Yusuff et al. [73], while the identification of Ni$^0$ was reported by Lima et al. [83] in AC samples produced with sapelli sawdust even after the application of activating agent leaching methods.

The average crystallite size (D) obtained by Scherer's equation for Ni and Cu particles was 30.70 nm and 37.00 nm, respectively. After sulfonation, the D of the Ni and Cu particles reduced to 24.30 nm and 27.30 nm. The decrease in D in metallic particles after sulfonation with H$_2$SO$_4$ was also observed in studies by Varão et al. [80] and Ibrarrim et al. [84].

After sulfonation, two diffraction peaks at 2θ ≈ 20–25° and 2θ ≈ 40°–45° corresponding to Braggs reflection planes of (002) and (100) were observed, respectively. The (002) plane is attributed to the random order of aromatic carbon sheets and the (100) plane to amorphous carbon [85,86]. The characteristic peaks of ZnO are not observed in the diffractogram of AC-S/ZnCl$_2$, indicating that H$_2$SO$_4$ promoted its complete leaching. In the diffractograms of AC-S/NiCl$_2$ and AC-S/CuCl$_2$, only a reduction in the intensity of the diffraction peaks of Ni$^0$ and Cu$^0$ is evident, suggesting partial leaching of these metals. These results corroborate the chemical composition data presented in Table 3. X-ray patterns of AC-S showed the characteristic peaks of the quartz (SiO$_2$, ICCD card 00–001–0649), possibly contained in the biomass [83] or arising from the glass flask wear used for the sulfonation.

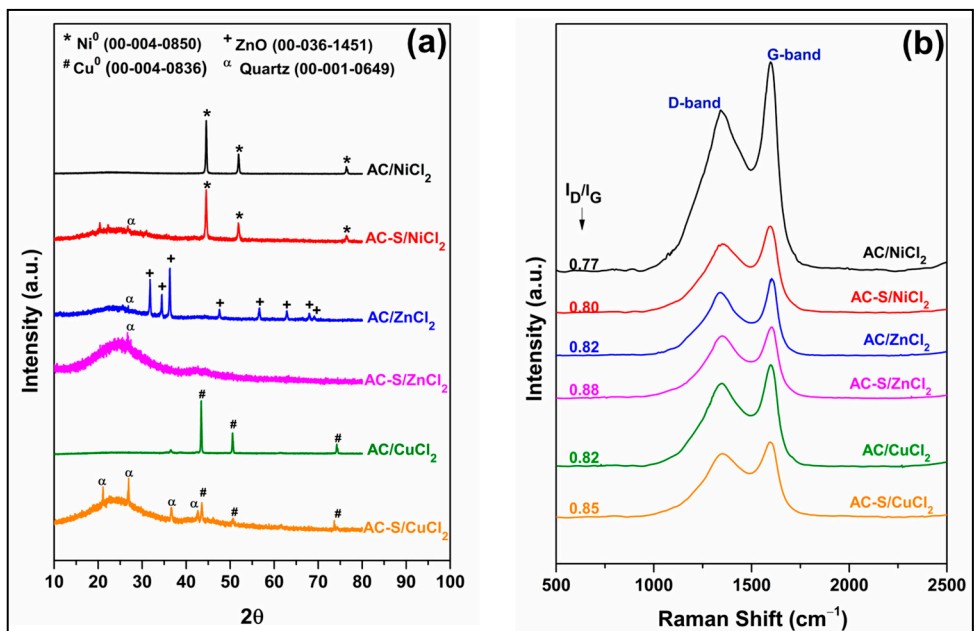

**Figure 4.** (**a**) XRD patterns and (**b**) Raman spectra for the ACs before and after sulfonation.

Raman spectra of the activated carbons before and after functionalized with $H_2SO_4$ are shown in Figure 4b. All samples display two broad peaks around 1350 cm$^{-1}$ (D-band) and 1590 cm$^{-1}$ (G-band). D-band is associated with amorphous and disordered carbon structure presenting in sp$_3$-hybridized carbon ($A_{1g}$ symmetry), while G-band is attributed to the $E_{2g}$ symmetry and is related to the C-C stretching bond on ring and chain sp$_2$- hybridized carbon [85,87,88]. The intensity ratios of the D-band and G-band ($I_D/I_G$–inserted in Figure 4b) indicate the disorder degree of the carbonic structure [85,87]. After sulfonation, all samples had a slight increase in the $I_D/I_G$ ratio that can be attributed to the formation of new defects of the carbonic matrix due to the incorporation of the -SO$_3$H group [89,90]. The slight increase in the $I_D/I_G$ observed in this study was also reported by Li et al. [91], Bounoukta et al. [85], and Pi et al. [92].

### 2.3. Conversion of Xylose into Furfural

The results of the xylose conversion (C%), yield (Y%), and selectivity (S%) of furfural at 180 °C/2 h for the different acid catalysts are systematized in Table 4.

The C% and Y% for the uncatalyzed reaction were lower than those obtained for the sulfonated catalysts. The C% for AC-S/ZnCl$_2$ and AC-S/CuCl$_2$ decreased by 18.60 and 13.30%, respectively, while the AC-S/NiCl$_2$ increased by 14.00%. Possibly, the reduction after sulfonation is related to the leaching of 100% of ZnO and 90% of Cu$^0$ (see Table 4), which act as Lewis acid sites in the xylose conversion. Thus, the xylose conversion by these catalysts started to be carried out mainly by the Brønsted acid sites represented by the -SO$_3$H group. In this perspective, the C% increase seen for CA-S/NiCl$_2$ can be attributed to the synergy between Lewis acid sites (Ni$^0$) [93] and Brønsted acid sites (-SO$_3$H) present on this catalyst. According to Gabriel et al. [20], problems related to lower conversion values can be minimized by recirculating and submitting the unconverted substrate to a new reactional cycle.

All sulfonated samples had improved yield and selectivity to furfural compared with the non-sulfonated catalysts, but AC-S/NiCl$_2$ and AC-S/CuCl$_2$ showed better catalytic performance. The best catalytic activity of these acid solids is related to the simultaneous presence of Lewis and Brønsted acid sites because the dehydration mechanism of xylose into furfural can occur directly by a Brønsted acid site or in a cascade when xylose initially is isomerized into xylulose and lyxose by a Lewis acid site, and then dehydrated in the presence of a Brønsted acid site, being the second mechanism most favorable for the

selective conversion of xylose to furfural [12]. The combination of Lewis and Brønsted acid sites in catalysts has been associated, in different studies available in the literature, with obtaining higher yield and selectivity values for the dehydration reaction from xylose to furfural [12,15,94–96].

Comparing the two acid catalysts with the best catalytic performance, it is observed that AC-S/CuCl$_2$ presented higher values of Y% and S% for the reaction studied. The lower catalytic performance obtained for AC-S/NiCl$_2$ may be associated with the formation of furan resins and condensation products generated by furfural degradation. The high selectivity obtained for AC-S/CuCl$_2$ can reduce the costs of the furfural production process because when the catalyst has poor selectivity, it is necessary to install units for separating and purifying the product [20].

In Table 5, a comparison is made between the AC-S/CuCl$_2$ catalyst and different heterogeneous acid catalysts reported in the literature.

**Table 5.** Comparison of AC-S/CuCl$_2$ with other heterogeneous acid catalysts used for furfural production.

| Catalyst | Solvent | T [a] (°C) | T [b] (min) | C [c] (%) | Y [d] (%) | S [e] (%) | Reference |
|---|---|---|---|---|---|---|---|
| AC-S/CuCl$_2$ | Water | 180 | 120 | 66.67 | 55.96 | 83.93 | This study |
| NbPO$_4$ | Water | 160 | 60 | 44.10 | 17.40 | 39.50 | [97] |
| Amberlyst 70 | Water | 160 | 100 | - | 18.00 | - | [98] |
| l-12Nb | Water/Toluene | 160 | 240 | 62.00 | 36.60 | 59.00 | [99] |
| Si-12Nb | Water/Toluene | 160 | 240 | 42.00 | 33.60 | 80.00 | |
| s-MWCNTs | Water | 170 | 180 | 62.70 | 35.80 | 57.10 | [27] |
| AC(Zn)/S | Water | 180 | 120 | ~20.00 | ~10.00 | 50.00 | [33] |
| Starbon®-SO$_3$H | Water/ Methoxycyclopentane | 150 | 750 | 78.50 | 54.40 | 69.20 | [50] |
| Starbon®-SO$_3$H | Water/ Methoxycyclopentane | 200 | 750 | 100 | 50.30 | 50.30 | |
| Starbon®-SO$_3$H | Water/ Methoxycyclopentane | 200 | 1140 | 100 | 21.00 | 21.00 | |
| Amberlyst 70 | Water | 200 | 960 | 71.80 | 28.80 | 40.11 | [100] |
| M-20 | Water | 200 | 480 | 52.50 | 25.40 | 48.40 | |
| M-20 | Water | 200 | 960 | 87.60 | 44.20 | 50.45 | |
| M-20 | Water/Toluene | 200 | 960 | 75.80 | 42.60 | 56.20 | |
| ZSM-5-30 | Water | 200 | 480 | 57.70 | 19.90 | 34.50 | |
| ZSM-5-30 | Water | 200 | 960 | 82.50 | 34.10 | 41.33 | |
| mNb-bc | Water | 140 | 120 | 41.20 | 31.80 | 77.10 | [20] |
| Nb$_2$O$_5$ | Water | 160 | 360 | 76.80 | 46.16 | 60.10 | [101] |
| Nb$_2$O$_5$ | Water/Isopropanol | 160 | 360 | 99.00 | 34.05 | 34.40 | |
| Nb$_2$O$_5$/Al$_2$O$_3$ | Water | 160 | 360 | 87.40 | 30.68 | 35.10 | |
| Nb$_2$O$_5$/Al$_2$O$_3$ | Water/Isopropanol | 160 | 360 | 91.10 | 20.22 | 22.20 | |
| OMC-SO$_3$H | Water | 200 | 45 | 8.90 | 2.50 | 28.08 | [102] |

[a] T—temperature. [b] t—time. [c] C (%)—xylose conversion. [d] R (%)—yield furfural. [e] S (%)—selectivity.

Among the catalysts produced in this study and compared with the other acid solids catalysts reported in Table 5, the AC-S/CuCl$_2$ proved to be a promising catalyst for furfural production from xylose since it reached the highest yield (55.96%) and selectivity (83.93%) values in an aqueous medium. This catalyst also showed better Y% and S% compared to

the heterogeneous catalysts used in biphasic systems reported in the studies by Millán et al. [50], Sato et al. [100], and Lima et al. [101].

## 3. Materials and Methods

### 3.1. Precursor Material

The raw sugarcane bagasse (SCB) used as a precursor to obtain activated carbon was provided by the Jatiboca Sugar and Ethanol Plant (Urucânia, Minas Gerais, Brazil). The material was previously dried in the sun until it reached a humidity level below 15%; then, it was stored in plastic bags at room temperature.

The SCB used in this study contained 48.70% cellulose, 21.14% hemicelluloses, 24.81% lignin, 2.31% extractives, and 0.41% ash [103].

### 3.2. Preparation of Activated Carbon: Impregnation and Carbonization

Activated carbons (ACs) were obtained by the chemical activation method, as described by Thue et al. [55] and Oliveira et al. [104]. To prepare the ACs, 10.0 g (dry basis) of the SCB was impregnated with 150 mL of aqueous solutions containing the following activating agents: $ZnCl_2$, $NiCl_2$, and $CuCl_2$, according to experimental conditions described in Table 6. Then, the material was placed on the heating plate under agitation to remove excess water and finally dried in an air circulation oven at 105 °C for 12 h.

**Table 6.** Preparation conditions of the ACs.

| Activating Agent | Biomass/Activating Agent Ratio (m:m) | Carbonization Temperature | Nomenclature of Activated Carbon |
|---|---|---|---|
| $NiCl_2 \cdot 6H_2O$ | 1:1 | 550 °C | $AC/NiCl_2$ |
| $ZnCl_2$ | 1:1 | 550 °C | $AC/ZnCl_2$ |
| $CuCl_2 \cdot 2H_2O$ | 1:1 | 550 °C | $AC/CuCl_2$ |

After the impregnation process, the precursor material was carbonized in a tubular oven with a quartz tube at a temperature of 550 °C for 3 h under nitrogen flow. The material resulting from the carbonization process (AC) was macerated, washed with hot water to unblock the pores, and dried in an air circulation oven at 105 °C for 12 h.

The gravimetric yield of the activated carbon production process was calculated by Equation (1).

$$Y_C = \frac{M_{AC}}{M_P} \times 100 \qquad (1)$$

where $Y_C$ represents the carbonization process yield of AC, $M_{AC}$ is the mass (dry base) of the AC after carbonization, and $M_P$ is the mass (dry base) of the precursor.

### 3.3. Activated Carbon Sulfonation Process

The sulfonation of AC was performed by direct reaction with concentrated sulfuric acid (98%) in a solid/liquid ratio of 1:10, a temperature of 160 °C, and a reaction time of 2 h. After the reaction time, the AC was washed with distilled water until neutral pH and dried in an air circulation oven at 105 °C for 12 h, being called AC-S. The yield of the sulfonation process was calculated according to Equation (2).

$$Y_S = \frac{M_{AC\text{-}S}}{M_{AC}} \times 100 \qquad (2)$$

where $Y_S$ is the yield of the sulfonation process, $M_{AC}$ is the mass of the AC before sulfonation (dry base), and $M_{AC\text{-}S}$ is the mass of the AC after sulfonation.

*3.4. Catalyst Characterization*

3.4.1. Textural Properties

The specific surface area, total volume pore, and pore average diameter of ACs before and after sulfonation were determined using Brunauer–Emmet–Teller (B.E.T.) and Barret–Joyner–Hallenda (B.J.H.) methods. Nitrogen adsorption/desorption isotherms at $-196\,°C$ were obtained using ASAP 2020 equipment (Micromeritics).

3.4.2. Elemental Analysis

Carbon (C), hydrogen (H), nitrogen (N), and sulfur (S) contents of AC samples were determined by elemental analysis (CHNS) in 2400 Series II CHNS/O equipment (Perkin Elmer). The oxygen content (O) was obtained by difference, considering the summation of C, H, N, S, Cl, Si, Ni, Zn, and Cu.

3.4.3. Ion Exchange Capacity Analysis

The ion exchange capacity (IEC) of CA and CA-S was determined by acid-base titration according to the procedure adapted from González et al. [105] and Dechakhumwat et al. [106]. For determination of the IEC, a mass between 0.1 g and 0.2 g of AC was added to 20 mL of NaCl solution (1.0 M) and kept under stirring for 24 h. Subsequently, the mixture was filtered and titrated in triplicate with standard NaOH solution (0.05 mol $L^{-1}$), using phenolphthalein as an indicator. The IEC was calculated according to Equation (3).

$$\text{IEC (mmol/g)} = \frac{V_{NaOH} \times M_{NaOH}}{m} \tag{3}$$

where $V_{NaOH}$ is the volume (L) of solution spent on titration, $M_{NaOH}$ is the solution molarity, and m(g) is the mass of the AC.

The density of the $SO_3H$ groups (mmol/g) incorporated in the AC after the sulfonation process was estimated by the difference between the IEC of the material before sulfonation and the IEC after functionalization, according to Equation (4).

$$SO_3H \text{ (mmol/g)} = IEC_2 - IEC_1 \tag{4}$$

where $IEC_1$ is the ion exchange capacity of AC before sulfonation, and $IEC_2$ is the ion exchange capacity of AC after sulfonation.

3.4.4. Scanning Electron Microscopy Analysis (SEM)

The AC morphology was obtained using a Scanning Electron Microscope, model EVO 10 MA (Carl Zeiss, Jena, FRG), with a magnification of 1.00 KX and 5.00 KX and an acceleration voltage of 20 kV. The samples were fixed on metallic supports with double-sided carbon tape and metalized with a 75 nm gold layer.

The composition of Ni, Zn, Cu, Si, and Cl in the samples before and after sulfonation was determined by X-ray energy-dispersive spectroscopy (EDX) using an Oxford detector, model 51-ADD0048 (Oxford Instruments, Abingdon, UK).

3.4.5. Thermogravimetric Analysis (TGA)

The evaluation of the thermal stability of the AC and AC-S in an inert atmosphere was performed on a Thermogravimetric Analyzer, model TGA 55 (TA Instruments, New Castle, USA). Approximately 6 mg of the samples were subjected to a heating ramp from 25 to 900 °C with a flow rate of 60 mL $min^{-1}$ of nitrogen and a heating rate of 10 °C $min^{-1}$.

3.4.6. X-ray Diffraction Analysis (XRD)

X-ray diffractograms of activated carbon after and before functionalization with sulfuric acid were obtained using an X-ray Diffractometer, model XRD-6000 (Shimadzu, Tokyo, Japan) with 10–80° (2θ) range, scanning speed 2°/min, power of 40 kV with electric current of 30 mA, and Cu-Kα radiation (λ = 1.5406 Å).

The average crystallite size of Ni and Cu particles in AC and AC-S were obtained, respectively, from the XRD patterns at $2\theta \approx 44.50°$ (111) and $2\theta \approx 43.29°$ (111) using Scherrer's equation (Equation (5)) [55].

$$D = 0.9\lambda/\beta \cos(\theta) \tag{5}$$

where D is the average crystallite size, $\lambda$ is the wavelength of the X-ray, $\theta$ is the angle obtained from $2\theta$ value corresponding to the XRD pattern, and $\beta$ is the full width at half maximum (FWHM).

### 3.4.7. Raman Spectroscopy Analysis

Raman scattering measurements were performed at room temperature in a Raman Spectrophotometer, model Labram HR Evolution (Horiba Scientific, Kyoto, Japan), equipped with a CCD detector. Raman spectra were obtained in the excitation line of 532 nm [107], spectral resolution of $0.01$ cm$^{-1}$, incidence power of 1.7 mW, and four accumulations were performed with a time of 20 s for each spectrum in the region of 800 to 2200 cm$^{-1}$. The structural defects of the AC and AC-S were determined by the relationship between intensities of the $I_D/I_G$ bands.

### 3.5. Application of Activated Carbon for Conversion of Xylose to Furfural

#### 3.5.1. Experimental Conditions

The reactions of the production of furfuraldehyde from the xylose dehydration were conducted in a Parr 4848 reactor (Parr Instrument Company, Moline, USA) with a capacity of 450 mL and equipped with a heating system, magnetic stirring, and temperature controller. The xylose dehydration tests were performed using a mass percentage of catalyst/xylose corresponding to 5%. To perform the test, 30 mL of an aqueous solution of xylose was added to the reactor along with 0.05 g of catalyst. The mixture was then bubbled with nitrogen gas ($N_2$) for 5 min to remove dissolved gases that could interfere with the dehydration reaction. Afterward, the reaction mixture was heated to 180 °C, with an average time to reach the working temperature of 65 min, and maintained at that temperature for 2 h. Reaction conditions were defined based on previous studies of Termividchakorn et al. [27] and Lin et al. [33].

Posteriorly, the mixture was filtered through quantitative filter paper to separate the activated carbon. The liquid fraction was filtered again using Millipore® filters (0.20μm) and then frozen for later determination of the xylose and furfural concentration.

#### 3.5.2. Determination of Xylose and Furfural

The xylose and furfural concentration were determined by high-performance liquid chromatography (HPLC) using a Shimadzu Chromatograph coupled to a refractive index detector (RID-10A), diode array detector (SPD-M20A) (Shimadzu, Tokyo, Japan), and Supelcogel C610 column-H (30 cm × 7.8 mm) (Sigma-Aldrich, Bellefonte, USA). The xylose separation was carried out at 32 °C using a RID-10A detector with an aqueous phosphoric acid solution (0.1% $v/v$) at a flow rate of 0.5 mL.min$^{-1}$, while the determination of furfural was performed at 55 °C in a PDA detector (274 nm) with an aqueous sulfuric acid solution (5 mmol L$^{-1}$) at a flow rate of 0.6 mL min$^{-1}$.

Xylose conversion (C%), yield (Y%), and selectivity (S%) furfural were determined, respectively, by Equations (6)–(8) [108].

$$C(\%) = \frac{[C_{xi}] - [C_{xf}]}{[C_{xi}]} \times 100 \tag{6}$$

$$Y(\%) = \frac{[C_F]}{[C_{xi}]} \times 100 \tag{7}$$

$$S(\%) = \frac{[\text{furfural yield}]}{[\text{xylose conversion}]} \times 100 \tag{8}$$

where $C_{xi}$ is the initial concentration of xylose (mol $L^{-1}$), $C_{xf}$ is the final concentration of xylose (mol $L^{-1}$), and $C_F$ is the concentration of furfural produced.

## 4. Conclusions

In this study, three AC-S were prepared from the sugarcane bagasse for application in the dehydration of xylose in furfural. To the best of our knowledge, it is the first reported approach that synthesized AC-S without performing the activating agent leaching step before direct functionalization with $H_2SO_4$, thus enabling the obtainment of single-step catalysts with Lewis and Brønsted acid sites represented, respectively, by the metals and the -$SO_3H$ group. The textural properties determined by BET and BJH methods showed that ACs activated using $ZnCl_2$ are classified as a microporous material with high $S_{BET}$ (1055 $m^2$ $g^{-1}$ to 1100 $m^2$ $g^{-1}$), while ACs produced with $NiCl_2$ and $CuCl_2$ have lower SBET (290 $m^2$ $g^{-1}$ to 440 $m^2$ $g^{-1}$) and are mesoporous materials. The insertion of -$SO_3H$ groups in the carbonic matrix by the direct sulfonation method was successfully performed, as evidenced in the techniques of EDX, TGA, elemental analysis, and ion exchange capacity. X-ray diffraction analysis showed that ACs have an amorphous structure. SEM-EDX and Raman analyses evidenced that the sulfonation process did not cause significant structural changes in the ACs.

Among the catalysts studied, the AC-S/$CuCl_2$ showed the best catalytic performance for furfural production in an aqueous medium at 180 °C/2 h, reaching a xylose conversion of 66.67%, yield and selectivity of 55.96% and 83.93%, respectively. Given the search for technologies and processes that convert lignocellulosic residues in a green and sustainable way into biofuels and value-added chemicals, the results of this study may contribute to the development of heterogeneous catalysts that provide the selective conversion of xylose into furfural, such as the AC-S/$CuCl_2$ produced in this work.

**Author Contributions:** T.A.L.S.: conceptualization, methodology, validation, formal analysis, investigation, writing—original draft, writing—review and editing. A.C.d.S.: conceptualization, methodology, formal analysis, resources, writing—original draft, writing—review and editing, supervision. D.P.: conceptualization, methodology, formal analysis, resources, writing—original draft, writing—review and editing, supervision, project administration, funding acquisition. All authors have read and agreed to the published version of the manuscript.

**Funding:** This study was financed in part by Fundação de Amparo à Pesquisa do Estado de Minas Gerais (FAPEMIG), Conselho Nacional de Desenvolvimento Científico e Tecnológico (CNPq), and Coordenação de Aperfeiçoamento de Pessoal de Nível Superior—Brazil (CAPES)—Finance Code 001.

**Data Availability Statement:** Not applicable.

**Acknowledgments:** The authors would like to thank the Rede de Laboratórios Multiusuários (RELAM/PROPP) at the Federal University of Uberlândia for providing the equipment and technical support for the experiments.

**Conflicts of Interest:** The authors declare no conflict of interest.

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
