# Peer review of "Synthesis and Characterization of Acid-Activated Carbon Prepared from Sugarcane Bagasse for Furfural Production in Aqueous Media"

_catalysts, doi:10.3390/catal13101372_

Round 1

Reviewer 1 Report

This manuscript reports on a simple preparation method of a solid acid catalyst, which catalyzes the production of furfural from xylose in aqueous solution, with excellent reaction efficiency. This finding is significant for advancing the application of biomass-derived platform compounds. Therefore, it is suggested that this manuscript be accepted, after the following questions are addressed:

1. The author notes in the introduction that references32-36 used a sulfonated carbon catalyst for the furfural production process from xylose, but the specific results are not detailed. What are the shortcomings of these references, and compared to them, what improvements does this manuscript offer?

2. Even after treatment with concentrated sulfuric acid and washing with water, there is still a considerable amount of Ni and Cu residue. The XRD spectrum also shows the presence of these Ni and Cu. It would be beneficial to provide particle size information for Ni and Cu, such as the particle sizes measured by XRD, chemical adsorption, and TEM.

3. In Figure 2, there are two sets of identical labels a-f, please change to different labeling methods to avoid confusion. In addition, the identification under SEM is too vague to see clearly, please revise.

Author Response

Dear reviewer, initially, we would like to thank you for the suggestions given to improve the manuscript “Synthesis and characterization of acid-activated carbon pre-pared from sugarcane bagasse for furfural production in aqueous media". In the attached file, you find all the answers to the comments performed.

Reviewer 2 Report

The work shows the performance of carbons prepared from the sugarcane bagasse for application in the catalytic dehydration of xylose in furfural. Before their sulfonation, the samples were activated. The work clearly shows the effect of the activating agent on the selectivity of the sugar dehydration reaction towards furfural.

The work is well-written, and the topic is of high interest. I suggest the acceptance of the manuscript after making some minor corrections.

Line 211. Please check the font size.

Figure 3. For a better visualization of the change in mass with temperature, it is suggested to show the derivative of the TGA curves.

Line 340 “the reduction of Ni2+, Zn2+, and Cu2+ to Ni0, ZnO, and Cu0.” Please revise. ZnO is an oxide.

The xylose conversion and selectivity towards furfural in the absence of catalysts should be shown.

What by-products were observed?

Could the authors quantify the percentage of Lewis and Bronsted sites?

Author Response

(The authors gave the same response as above.)

Reviewer 3 Report

In the present work, several sulfonated activated carbons, prepared by different chemical activation processes, have been synthesized, characterized and evaluated as solid acid catalysts for the production of furfural from xylose. Although the topic is attracting the interest of many research groups and the catalytic results are very good (56% furfural yield at 180ºC, after 2 h, in water), there are several issues that require to be addressed to clarify the information provided in the manuscript:

1. The number of references should be reduced (there are 106), including only those most directly related to the present work. For instance, there are many citations about the use of activated carbons for biodiesel production through esterification of free fatty acids. On the other hand, considering that activated carbons have also been proposed as catalysts for xylose dehydration to furfural, it is necessary to point out no only their advantages, but also their main drawbacks which drive the search for new synthesis protocols.

2. The highest BET surface area of activated carbon obtained by chemical activation with ZnCl2, compared to the rest of the catalysts, is explained due to <the swelling of the molecular structure of cellulose, which causes the breaking of its lateral bonds, and promotes the increase of intra- and inter fibril spaces>. Why is this phenomenon not observed when nickel and copper chlorides are used?

3. The lower sulfonation yield of AC/CuCl2 when compared to AC/NiCl2 and AC/ZnCl2, is justified by the <reaction of copper ions and H2SO4, which are strong oxidizing agents, with the loose carbon atoms in macro and mesopores of the carbonic structure initially formed. In addition, this process leads to an increase in the BET surface area and causes the degradation of the carbonic matrix and consequently the reduction of yield>. However, the AC-S/CuCl2 catalyst, according to data shown in Table 4, has the highest carbon content (63.71 %), and copper is present, according to the XRD data, as metal copper nanoparticles, and not as copper ions. A more convincing explanation is required.

4.   How is it explained that the treatment with concentrated sulfuric acid, in a solid:liquid ratio of 1:10, at 160ºC for two hours, leaches all the Zn, almost all the copper (ca. 90 wt%), but leaves 66 wt% of the nickel without leaching, when the latter metal is more oxidizable than copper?

5.   It is very difficult to accept the existence of metallic copper and nickel after treatment with a highly oxidizing acid, concentrated sulfuric acid, at high temperature (160ºC), in water.

6.   Why do the diffraction signals associated with quartz become more intense and narrow after the sulfonation process of the AC/CuCl2 sample? The silicon content must be reflected in the elemental chemical composition data in Table 4.

7.  The authors conclude that the results obtained show the importance of the synergy between the Lewis and Brønsted acid sites in the selective dehydration of xylose. However, in the present work, the nature of the acid sites, nor their strength and concentration, are not studied. Furthermore, Ni(0) is claimed to be associated with Lewis acid sites, but this is not supported by any experimental studies, as these are more metallic than acidic sites.

8.  The catalytic data (Table 5) must be explained considering the textural and acid properties of the different materials (sulfonated and non-sulfonated), and this is not done in the present work. For example, how do they explain the high furfural yield of the AC-S/ZnCl2 catalyst (31.27%), with 0.05 mmol/g of acid sites and where all the Zn has been leached?

9.  One of the main advantages of heterogeneous catalysis, compared to the homogeneous one, is the possibility of recovering the used catalyst and being able to reuse it in new catalytic cycles, to evaluate its activity, and, consequently, its stability. This study is mandatory to be able to accept this work, since the dehydration process of xylose to furfural could be catalyzed by the soluble species of the catalyst (it should not be forgotten that water at 180ºC favors hydrolysis reactions in solids).

10. X-ray diffractogram patterns, chemical composition, and textural data of used catalysts should be included to assess their stability in the reaction medium.

Therefore, all these points must be clarified in order to recommend this manuscript for publication in Catalysts

Author Response

(The authors gave the same response as above.)

Round 2

Reviewer 3 Report

The authors have endeavoured to adequately respond to the questions raised in the review process. In most cases, this has allowed the information provided in the manuscript to be improved and clarified, and according to the other reviewers, it could be recommended for publication. However, there are key issues regarding the determination of acidic properties of catalysts and their reusing that are not addressed due to lack of material for testing.